# Efficient plant genome engineering using a probiotic sourced CRISPR-Cas9 system

Zhaohui Zhong[1,2,10], Guanqing Liu [3,4,5,10], Zhongjie Tang[1,10], Shuyue Xiang[1,10], Liang Yang[6,7], Lan Huang[1], Yao He[1], Tingting Fan [1], Shishi Liu[1], Xuelian Zheng [1,2], Tao Zhang [3,4,5], Yiping Qi [8,9] ✉, Jian Huang [1] ✉ & Yong Zhang [1,2] ✉

Among CRISPR-Cas genome editing systems, *Streptococcus pyogenes* Cas9 (SpCas9), sourced from a human pathogen, is the most widely used. Here, through in silico data mining, we have established an efficient plant genome engineering system using CRISPR-Cas9 from probiotic *Lactobacillus rhamnosus*. We have confirmed the predicted 5′-NGAAA-3′ PAM via a bacterial PAM depletion assay and showcased its exceptional editing efficiency in rice, wheat, tomato, and Larix cells, surpassing LbCas12a, SpCas9-NG, and SpRY when targeting the identical sequences. In stable rice lines, LrCas9 facilitates multiplexed gene knockout through coding sequence editing and achieves gene knockdown via targeted promoter deletion, demonstrating high specificity. We have also developed LrCas9-derived cytosine and adenine base editors, expanding base editing capabilities. Finally, by harnessing LrCas9's A/T-rich PAM targeting preference, we have created efficient CRISPR interference and activation systems in plants. Together, our work establishes CRISPR-LrCas9 as an efficient and user-friendly genome engineering tool for diverse applications in crops and beyond.

The CRISPR-Cas system, an innate immune system in many bacteria and archaea, has been harnessed to develop genome engineering tools. Since the first demonstration of CRISPR-Cas9 for inducing targeted DNA double-strand breaks (DSBs) in prokaryote and eukaryote cells[1,2], this powerful tool has revolutionized genome engineering including in plants[3–7]. *Streptococcus pyogenes* Cas9 (SpCas9), sourced from a human pathogen, has become the most widely used Cas9.

And it has been adopted to achieve versatile plant genome engineering including base editing[8,9], prime editing[10,11], gene regulation[12–14], and epigenetic editing[15,16].

SpCas9 mainly recognizes a 5′-NGG-3′ PAM, which limits its targeting scope in the genomes. Two approaches have been adopted to overcome this problem. In the first approach, researchers focused on the discovery of functionally useful Cas9 orthologs with alternative

[1]Department of Biotechnology, School of Life Sciences and Technology, Center for Informational Biology, University of Electronic Science and Technology of China, 610054 Chengdu, China. [2]Chongqing Key Laboratory of Plant Resource Conservation and Germplasm Innovation, Integrative Science Center of Germplasm Creation in Western China (Chongqing) Science City, School of Life Sciences, Southwest University, 400715 Chongqing, China. [3]Jiangsu Key Laboratory of Crop Genomics and Molecular Breeding/Jiangsu Key Laboratory of Crop Genetics and Physiology, Agricultural College of Yangzhou University, 225012 Yangzhou, China. [4]Key Laboratory of Plant Functional Genomics of the Ministry of Education/Joint International Research Laboratory of Agriculture and Agri-Product Safety, The Ministry of Education of China, Yangzhou University, 225012 Yangzhou, China. [5]Jiangsu Co-Innovation Center for Modern Production Technology of Grain Crops, Yangzhou University, 225012 Yangzhou, China. [6]Horticulture Research Institute, Sichuan Academy of Agricultural Sciences, Sichuan, China. [7]Vegetable Germplasm Innovation and Variety Improvement Key Laboratory of Sichuan Province, 610066 Chengdu, China. [8]Department of Plant Science and Landscape Architecture, University of Maryland, College Park, MD 20742, USA. [9]Institute for Bioscience and Biotechnology Research, University of Maryland, Rockville, MD 20850, USA. [10]These authors contributed equally: Zhaohui Zhong, Guanqing Liu, Zhongjie Tang, Shuyue Xiang. ✉e-mail: Yiping@umd.edu; hj@uestc.edu.cn; zhangyong916@uestc.edu.cn

PAM requirements such as SaCas9[17], St1Cas9[18], NmeCas9[19], Nme2Cas9[20], CjCas9[21], and BlatCas9[22]. In general, these Cas9 orthologs recognize more complex PAM sequences with multiple nucleotides (6–8 bp) and possess limited editing activity. Recently, Cas9 orthologs with shorter PAMs have been reported such as ScCas9 which recognizes 5′-NNG-3′PAM[23], and FrCas9 which recognizes 5′-NNTA-3′PAM[24]. ScCas9 was demonstrated for genome editing in rice, albeit with low efficiency[25]. The second approach applies rational design or direct protein evolution to engineer SpCas9 variants with altered PAM requirements, such as xCas9 for a 5′-NG-3′ PAM[26], SpCas9-NG for a 5′-NG-3′ PAM[27], and SpRY for 5′-NR/YN-3′ PAMs[28]. In plants, xCas9 was proved to mainly recognize the canonical 5′-NGG-3′ PAM[29,30]. SpCas9-NG and SpRY indeed expand targeting scope by recognizing relaxed PAMs but the editing efficiency is low[29,31,32]. Such compromised editing efficiency could be partly attributed to their self-cleavage effects[9,31,33].

As an effective complement of the CRISPR-Cas9 system, the CRISPR-Cas12a system confers genome editing at canonical 5′-TTTV-3′ PAM sites, with LbCas12a being most widely used in plants[34–36]. While CRISPR-Cas12a is advantageous for multiplexed genome editing[37,38] and promoter editing[39], this system is sensitive to temperature[40]. Also, the lack of Cas12a nickases has hindered the development of more sophisticated precise genome editing tools such as base editors and prime editors. Only recently, efficient base editors using dCas12a proteins were reported for plant genome editing[41,42]. Although Cas12a-based CRISPRi systems were reported[34], potent Cas12a-based CRISPRa systems are yet to be developed. So, despite great progress, CRISPR-Cas12a has not matched the versatility of CRISPR-Cas9 as a genome engineering platform yet.

In this study, we report a type II-A Cas9 from probiotic *Lactobacillus rhamnosus GG* named LrCas9. LrCas9 recognizes a 5′-NGAAA-3′ PAM, which overlaps (as a reverse complementary strand) the 5′-TTTV-3′ PAM of Cas12a. LrCas9 is efficient for genome editing in multiple plant species such as rice, wheat, tomato, and larix with high-fidelity. Moreover, LrCas9 shows higher editing efficiency than SpCas9-NG, SpRY, and LbCas12a when targeting the same DNA sequences. Further, LrCas9-derived cytosine base editor (CBE) and adenine base editor (ABE) are developed and demonstrated in rice and wheat. Finally, we engineer CRISPRi and CRISPRa systems using LrCas9, and these systems enable potent transcriptional repression and activation in plant cells, respectively.

## Results

### *In-silico* mining of CRISPR-Cas9 systems de novo

To mine potentially useful CRISPR-Cas9 systems for genome engineering in plants, we took three steps that include Cas protein identification, CRISPR array identification, and PAM prediction (Fig. 1a). For Cas protein identification, we analyzed a total number of 33,825 proteomes, including 32,023 bacterium and 1832 archaea, and identified 30,495 CRISPR clusters: 29,586 from bacteria and 909 from archaea (Supplementary Fig. 1a). Among them, different types of CRISPR-associated protein were identified (Supplementary Fig. 1b). For CRISPR array identification, 4963 clusters that contain Type II Cas9 proteins were identified by using the CRISPRFinder[43]. We first identified the potential crRNA sequences for each cluster, and then the tracrRNA was each identified by crRNA sequence alignment. These analyses revealed the CRISPR locus lengths of 5000–10,000 bp, and predominantly ~7000 bp (Supplementary Fig. 1c). The Cas9 protein lengths span from 1000aa to 1400aa, but mainly ~1400aa (Supplementary Fig. 1d). The tracrRNA lengths range from 100 nt to 500 nt (Supplementary Fig. 1e) while the crRNA lengths are around 36 nt (Supplementary Fig. 1f). There were 127 CRISPR arrays that could align to phage genomes among 739 candidates (Supplementary Data 1). By identifying the anti-protospacers (anti-spacers) in virus/phage genomes using BLAST[44], we extracted the flanking sequences of anti-spacers and drew the PAMs for these 739 candidates using the WebLogo tool[45]. After removal of the redundant candidates and selection of CRISPR systems with high PAM scores, we narrowed down to 42 CRISPR-Cas9 candidates including many Type II-A systems and some Type II-C systems with different evolutionary lineages (Fig. 1b). There is structure difference between these two CRISPR-Cas9 types: Type II-A CRISPR-Cas9 loci consist of Cas9, Cas1, Cas2 and Csn2, while Type II-C CRISPR-Cas9 loci lack of Csn2 (Fig. 1b and Supplementary Figs. 2 and 3). A diverse panel of PAMs was predicted for the 42 candidate Cas9 nucleases (Supplementary Figs. 4 and 5). The crRNA of 42 candidates were aligned (Supplementary Fig. 6a), which revealed a conserved 5′-GUUUU-3′ motif at the 5′ end and 5′-AAAAC-3′ motif at the 3′ end (Supplementary Fig. 6b).

For experimental testing, we selected 2 Cas9 (BAI42646.1 from *Lactobacillus rhamnosus GG* and CAL43592.1 from *Flavobacterium psychrophilum JIPO2*) with the potential to broaden the PAM recognition, and 2 Cas9 (EPW83356.1 from *Streptococcus agalactiae STIR-CD-14* and OFH73969.1 from *Listeria monocytogenes*) that recognize the same PAM with SpCas9 to test whether any of them possesses higher nuclease activity. All four Cas9 proteins have a similar size to SpCas9 and contain all functional domains (Fig. 1c and Supplementary Fig. 7a–c). For example, The BAI42646.1 (named LrCas9) has the catalytic centers of RuvCI (D13) and HNH (H858) corresponding to D10 and H840 of the two domains in SpCas9 (Fig. 1c). The protein structures of SpCas9 and four Cas9 proteins mined from our dataset were predicted using SWISS-MODEL[46] (Fig. 1d and Supplementary Fig. 7d–f). Interestingly, the HNH domain of LrCas9 is located in the center of the protein, while the SpCas9's HNH domain faces backward from the center (Fig. 1d and Supplementary Fig. 8). This suggests different levels of conformation changes upon both Cas9 proteins' binding to tracrRNA-crRNA and target DNA. Based on PAM prediction, LrCas9 recognizes a 5′-NGAAA-3′ PAM (Fig. 1e), CAL4359.2 recognizes a 5′-NAATAT-3′ PAM, while EPW83356.1 and OFH73969.1 recognize a 5′-NGG-3′ PAM (Supplementary Fig. 7g–i). We selected rice endogenous sites to test these Cas9s' editing ability via co-transformation of protoplasts with all-in-one vectors expression crRNA, tracrRNA, and Cas9 (Fig. 1f). LrCas9 showed efficient genome editing with mutation rates from 4.3% to 64.2% at 3 selected sites (Fig. 1g), while other three Cas9 system showed no editing efficiency (Supplementary Fig. 7j–l). We thus focused on LrCas9 for further testing, especially because *Lactobacillus rhamnosus GG* is a probiotic strain.

### Characterization and optimization of CRISPR-LrCas9 for genome editing

Phylogenetic analysis showed that LrCas9 is closely related to SpCas9, ScCas9, and FrCas9, all belonging to the Type II-A Cas9 group (Fig. 2a). Based on sequence analysis, we confirmed the CRISPR locus in the genome of *Lactobacillus rhamnosus GG* (Fig. 2b). The coding sequence of this locus contains *Cas9*, *Cas1*, *Cas2*, and *Csn2*. The non-coding sequence contains a 138 bp tracrRNA sequence between *Cas9* and *Cas1*, and a CRISPR array composed of 23 spacer-crRNA units (Fig. 2b and Supplementary Data 2). We further validated the expression of tracrRNA and crRNA from small RNA-seq in *E. coli* that was transformed with this LrCas9 locus (Fig. 2b). To assess the PAM preference, we employed a depletion assay based on a double-antibiotics selection system in *E. coli* (Fig. 2c). The depletion assay showed the LrCas9 favored a 5′-NGAAA-3′ PAM (N = A, C, G, T) (Fig. 2d), which is consistent with the PAM prediction (Fig. 1e) and our initial genome editing results (Fig. 1g).

Our initial testing of the CRISPR-LrCas9 system used separate crRNA and tracrRNA (Fig. 1f). To combine both into a single guide RNA (sgRNA), we started with a long tetraloop (sgRNA V3.0) and made two truncated versions (sgRNA V2.0 and sgRNA V1.0) (Fig. 2e). We tested these three sgRNA scaffolds with 20 bp spacers at two independent 5′-NGAAA-3′ PAM sites (Os-CG01 and Os-TG01) in rice protoplasts. Both sgRNA V2.0 and V3.0 worked equally well (with ~20% to 25% editing

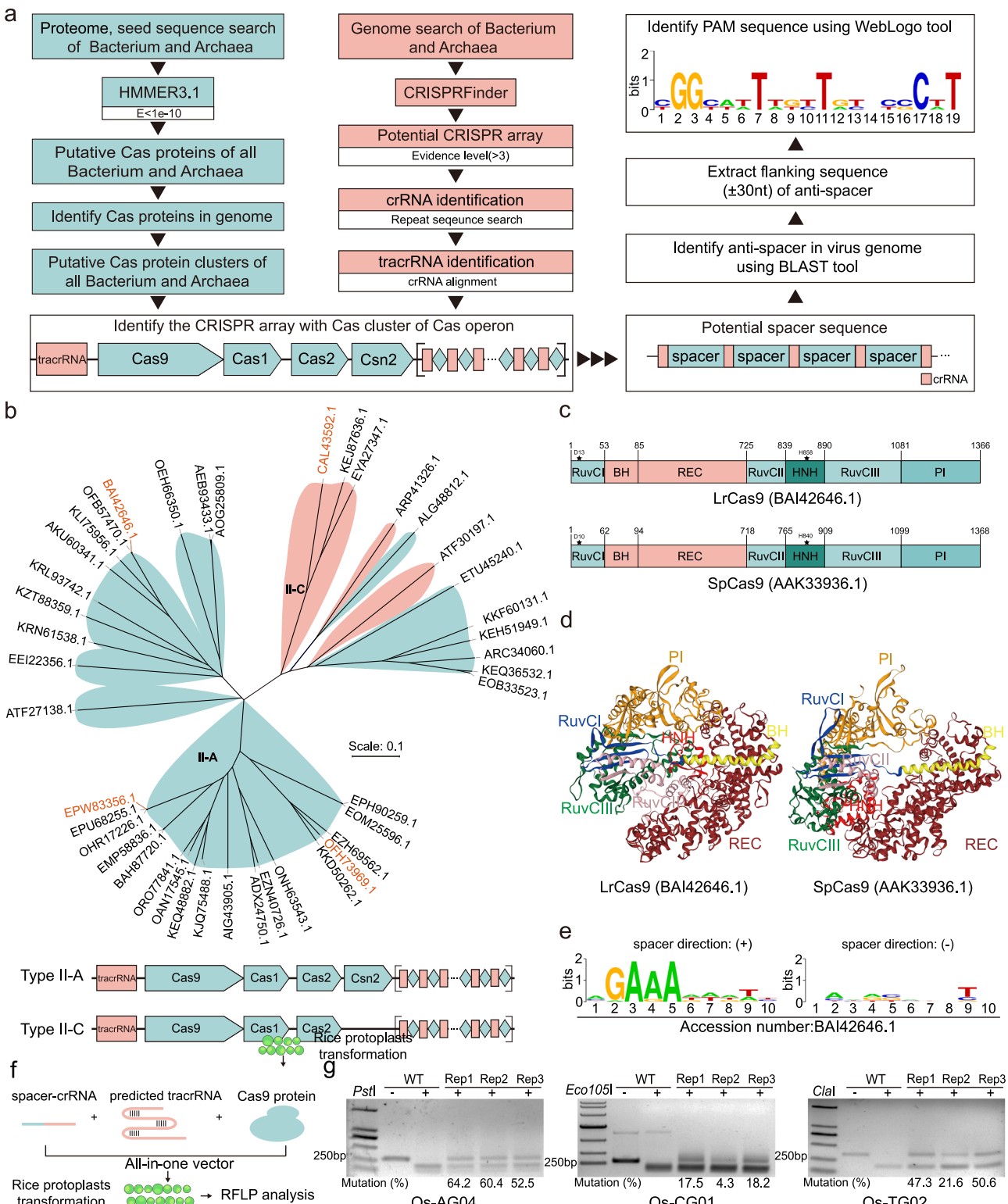

**Fig. 1 | Identification of de novo CRISPR-Cas systems for genome editing. a** Data-mining pipeline of de novo CRISPR-Cas systems. Box filled with cyan, Cas protein identification workflow; Box filled with salmon, CRISPR array identification workflow; Box filled in white, PAM prediction workflow. The CRISPR array was separated by square brackets. In the CRISPR array, the box in salmon indicates the crRNA while the diamond in cyan indicates the protospacer (spacer). The prediction results were listed in Supplementary Data 1. **b** Phylogenetic tree of 42 candidates in this study. The four further tested candidates were featured with salmon. The Type II-A systems were filled with cyan, and the Type II-C systems were filled with salmon. The structures of these two CRISPR systems are shown below. **c** The structural comparison of LrCas9 and SpCas9, and catalytic residues of RuvC and HNH domains of these two nucleases were shown with a pentagram. BH Bridge helix domain, PI PAM-interacting domain. **d** Predicted protein structures of LrCas9 and SpCas9 by SWISS-MODEL. **e** Predicted PAM of LrCas9 with different spacer's direction. (+), 5′−3′ direction. (−), 3′−5′ direction. **f** Editing ability test workflow of currently identified Cas9 nucleases in rice protoplasts. See details in the "Methods" section. **g** RFLP analysis results of LrCas9 at rice endogenous sites. The name of the restriction enzyme (RE) was shown at the top left of each gel. The mutation frequencies (proportions of uncut bands) were quantified by ImageJ software. Each experiment was repeated 3 times independently with similar results. Source data are provided as a Source Data file.

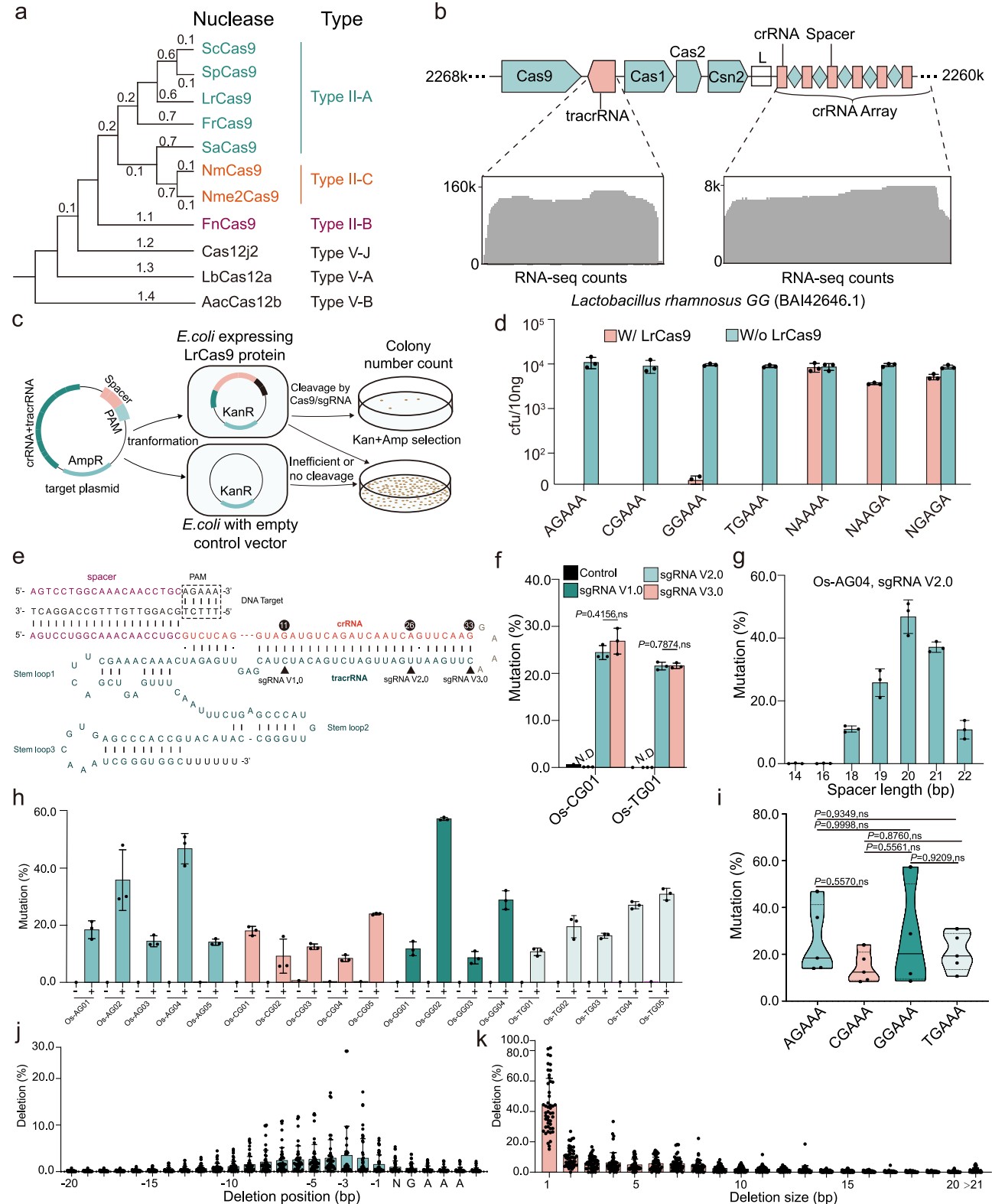

efficiency measured by next-generation sequencing (NGS)), while sgRNA V1.0 failed to work (Fig. 2f). Since sgRNA V2.0 is shorter than sgRNA V3.0, we selected the compact sgRNA V2.0 for further applications. We next assessed different spacer lengths (14–22 bp) by targeting the Os-AG04 site with sgRNA V2.0 and identified 20 bp as the optimal spacer length (Fig. 2g).

Having optimized CRISPR-LrCas9 with its PAM requirement, sgRNA scaffold, and spacer length, we wanted to evaluate the reliability

of this genome editing system. To this end, we targeted 19 endogenous sites in the rice genome and assessed editing efficiency using the protoplast system and NGS. Remarkably, LrCas9 showed detectable editing activity at all target sites, with efficiency ranging from 8.6% to 57.3% and averaged at 26.6% (Fig. 2h). Collective analysis of the editing data further confirmed that LrCas9 can edit all 5′-NGAAA-3′ PAM sites and the editing efficiencies at different 5′-NGAAA-3′ PAM groups showed no significant difference (Fig. 2i). By analyzing the genome

**Fig. 2 | Characterization and optimization of CRISPR-LrCas9 for genome editing. a** The phylogenetic tree of LrCas9 and other widely used CRISPR-Cas systems. The Type II-A systems were featured with cyan; the Type II-B systems were featured with purple, and the Type II-C systems were featured with salmon. **b** The CRISPR cluster structure in *Lactobacillus rhamnosus* GG (top), and the RNA-seq result of tracrRNA and crRNA array (bottom). **c** The LrCas9 PAM depletion assay workflow. See details in the "Methods" section. **d** The colony counts in the PAM depletion assay, indicating LrCas9 was efficient at NGAAA PAM sites. Each dot represents a biological replicate. Each target contains three independent experiments ($n = 3$). Data are presented as mean values ± SD. **e** The single guide RNA scaffold structure of LrCas9. The protospacer (spacer) was labeled in purple, the crRNA was labeled in salmon, the RNA linker GAAA was labeled in gray, and the tracrRNA was labeled in cyan. The different tetraloop lengths were pointed by triangles and named sgRNA V1.0, sgRNA V2.0, and sgRNA V3.0. The lengths of complementary regions were noted above the crRNA sequence with circles. **f** The mutation rates conferred by different sgRNA structures in rice protoplasts. ND not detectable, ns not significant.

Data were analyzed by two-way ANOVA multiple comparisons test with the Holm–Šídák method. ns, $P > 0.05$. Each design contains three independent experiments ($n = 3$). Data are presented as mean values ± SD. **g** The mutation rates by LrCas9 under different spacer lengths at Os-AG04 site with sgRNA V2.0. Each spacer length contains three independent experiments ($n = 3$). Data are presented as mean values ± SD. **h** The mutation rates of 19 endogenous sites in rice protoplasts by LrCas9. Each dot represents a biological replicate with an independent experiment. Each target contains three biological replicates ($n = 3$). Data are presented as mean values ± SD. **i** The summarized violin plot of mutation rate at different PAM groups of (**h**). The data were analyzed using two-way ANOVA with Tukey's multiple comparisons test. ns, $P > 0.05$. Each dot represents the average mutation rate of each target with three replicates. Solid line, median. Dash line, quartiles. **j–h** The deletion positions (**j**) and the deletion sizes (**k**) of all sites in (**h**). Each dot represents a biological replicate. Data are presented as mean values ± SD. Source data are provided as a Source Data file.

editing outcomes among all these sites, it appears that LrCas9 cut 3 bp upstream of the PAM (Fig. 2j), resulting in predominantly 1 bp deletions (Fig. 2k). Furthermore, the insertion and deletion proportions were also calculated, and LrCas9 caused more insertions than deletions at most target sites (Supplementary Fig. 9).

Previously, we found SpCas9 showed an optimal editing efficiency at 32 °C, where editing efficiency at 22 and 28 °C was reduced in rice protoplasts[40]. To assess the temperature sensitivity of LrCas9, we conducted a similar protoplast assay by editing the Os-GG02 site. Based on the result, LrCas9 showed comparable genome editing efficiency at 28 and 32 °C, whereas the editing efficiency was reduced at 22 °C (Supplementary Fig. 10). This data suggests that LrCas9 may be less temperature sensitive than SpCas9, making the CRISPR-LrCas9 a potentially robust genome editing system in plants.

## Benchmarking LrCas9 with other Cas nucleases and in diverse plant species

LrCas9 recognizes a 5′-NGAAA-3′ PAM, and its reverse complementary sequence forms a PAM for Cas12a nucleases such as the widely used LbCas12a (with a 5′-TTTV-3′ PAM) (Fig. 3a). In addition, SpCas9-NG and SpRY could recognize 5′-NGA-3′ PAM (Fig. 3b). These features allow for a close comparison between LrCas9 and these well-established Cas nucleases. We first compared the editing efficiency of LrCas9 and LbCas12a and its high-activity variants ttLbCas12a[47] and LbCas12a-RRV[48] at four independent loci in the rice genome[49,50] (Fig. 3a). Consistent with our recent report[48], the rice protoplast results showed that LbCas12a-RRV had the highest editing efficiency among Cas12a nucleases tested (Fig. 3c). Remarkably, LrCas9 had similar editing efficiency with LbCas12a at one site, but outperformed LbCas12a at the three other sites (Fig. 3c). Overall, LrCas9 is a significantly more potent nuclease than LbCas12a, with comparable editing efficiency to the ttLbCas12a variant (Fig. 3c). We also compared LrCas9 with SpCas9-NG and SpRY at editing seven independent sites with a 5′-NGA-3′ PAM (Fig. 3b). The protoplast assay showed that LrCas9 had significantly higher editing efficiency than SpRY and SpCas9-NG at five out of seven target sites, whereas all three nucleases failed to show detectable editing activity at two other sites (Fig. 3d). Together, these results demonstrate that LrCas9 is a more potent nuclease than LbCas12a, SpCas9-NG, and SpRY when targeting the same sequences with compatible PAM requirements.

LrCas9's tolerance to lower temperatures and higher nuclease activity discovered in rice made it a promising tool for genome editing in other plant species. To this end, we tested LrCas9 for editing seven target sites in wheat (a polyploid monocot) (Fig. 3e), seven target sites in tomato (a dicot) (Fig. 3f), and six target sites in Dahurian larch (also known as *larix gmelinii*, a coniferous tree) (Fig. 3g). Using protoplast assays, we found LrCas9 displayed editing activity at all target sites, with an average mutation rate of ~5% across these three species

(Fig. 3e–g). Such editing efficiencies are comparable to those previously reported for SpCas9 in wheat protoplasts[51] and in Dahurian larch protoplasts[52]. Hence, LrCas9 appears to be a robust nuclease among many plant species.

## Singular and multiplexed genome editing in stable rice plants
To test LrCas9 for genome editing in stably transgenic plants, we used six singular sgRNAs that target *OsPDS, OsDEP1, OsBADH2, Os03g0568400,* and *Os03g0603100* in rice (Supplementary Table 1). All six target sites were edited in T0 lines, with editing efficiency ranging from 16.7% to 85% and biallelic editing efficiency ranging from 10% to 40% (Supplementary Table 1). The resulting mutations were predominantly 1 bp insertion and a few bp deletions (Supplementary Fig. 11), consistent with the editing profile in rice protoplasts (Fig. 2k). These data support LrCas9 as an efficient genome editing tool for generating edited plants.

We next developed a multiplexed CRISPR-LrCas9 system using the tRNA-based sgRNA processing system[53]. We first multiplexed two sgRNAs to simultaneously edit *OsPDS* and *OsDEP1*. Testing the construct in rice protoplasts showed high editing efficiency (>20%) at both genes (Supplementary Fig. 12). Twenty-three T0 lines were generated for analysis. For *OsPDS*, editing efficiency and biallelic editing efficiency were 30.4% and 13%, respectively. For *OsDEP1*, editing efficiency and biallelic editing efficiency were 52.2% and 47.8%, respectively (Table 1). Moreover, for all seven T0 lines that *OsPDS* was edited, *OsDEP1* was also edited (Fig. 4a). Simultaneous editing could lead to chromosomal deletions. To test this, we multiplexed two sgRNAs (Os-TG01 and Os-TG02) for targeting *OsPDS* and generated 27 T0 lines. At Os-TG01 site, editing efficiency and biallelic editing efficiency were 85.2% and 81.5%, respectively. At the Os-TG02 site, editing efficiency and biallelic editing efficiency were 70.4% and 29.6%, respectively. For the lines that carry edits at Os-TG02, they all carry edits at Os-TG01 (Fig. 4b and Supplementary Fig. 13), which again suggests high-frequency simultaneous mutagenesis. Indeed, we were able to recover 4 T0 lines (14.8%) that contain large deletions due to simultaneous DSBs (Table 1). As expected, biallelic mutants of *OsPDS* showed an albino phenotype (Fig. 4c).

High-efficiency chromosomal deletion by LrCas9 further encouraged us to explore it for engineering quantitative traits via promoter editing as previously demonstrated in tomato[54], maize[55], and rice[56]. Recently, we developed a Cas12a promoter editing (CAPE) system that relies on target region prediction and efficient editing by Cas12a[39]. Since LrCas9 is a potent nuclease that is also very suitable for targeting A/T-rich promoters, we reasoned that LrCas9 would be a great tool for promoter editing. Previously, the promoter of *OsWx* (also known as *OsGBSS1*) has been edited for generating rice seeds with reduced amylose content. As a proof-of-concept, we designed a multiplexed LrCas9 construct for targeted deletion of ~2.1 kb in the *OsWx*

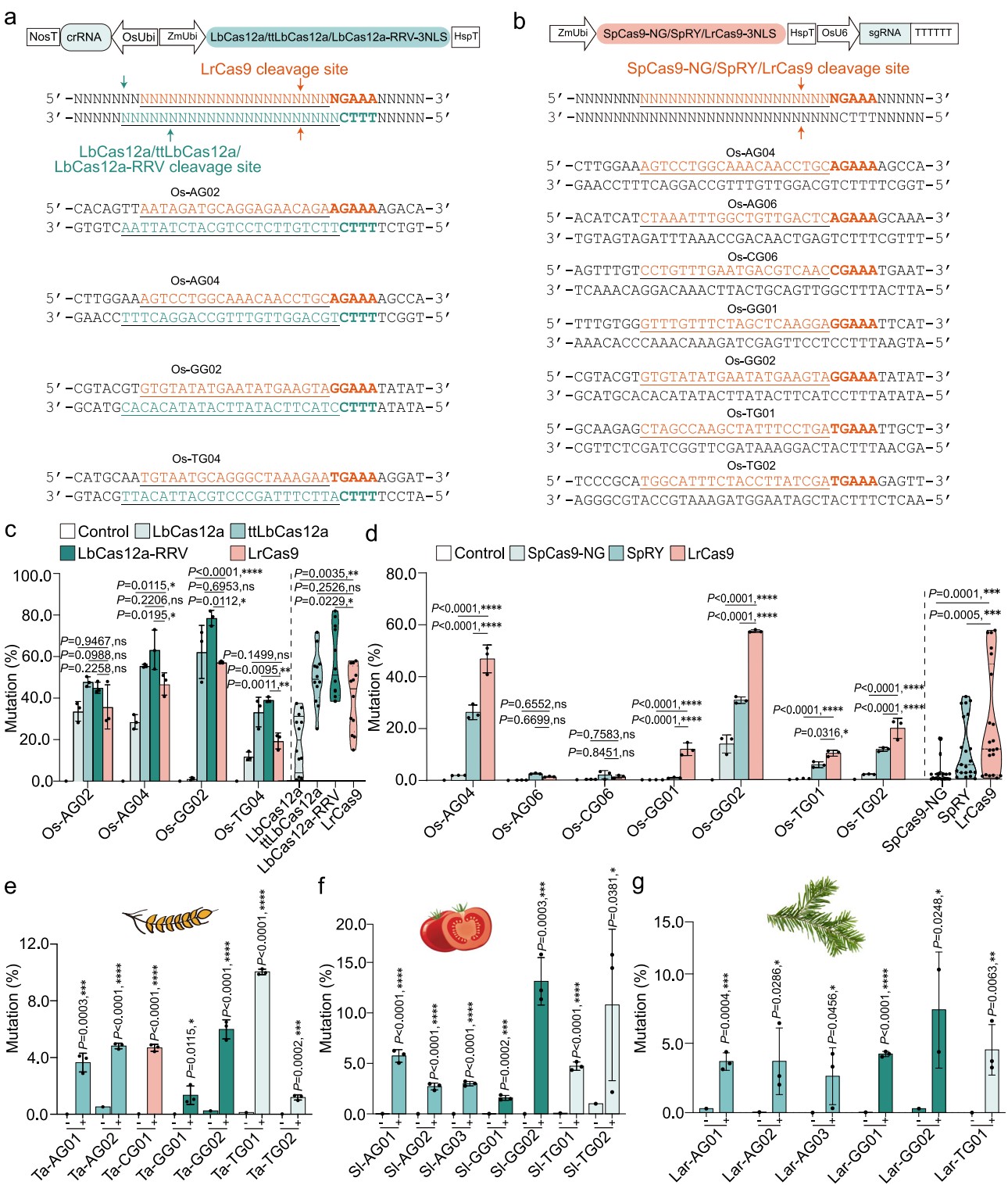

promoter, which contains key regions A–F (Fig. 4d). Based on our prediction, deletion of this region could significantly reduce *OsWx* expression. Screen of $T_0$ lines recovered multiple lines with such chromosomal deletions, and one line (no. 232-6) appears to be a homozygous deletion line (Fig. 4e and f). Transgene-free $T_1$ homozygous deletion lines were obtained from this $T_0$ line for further analysis. RNA expression of *OsWx* was reduced to ~40% of wild type (Fig. 4g). Accordingly, there was a 39.3% of reduction of amylose (Fig. 4h) and only an 8.1% reduction of total starch (Fig. 4i). Significant reduction of amylose content resulted in visible phenotypes of seeds before and after iodine staining (Fig. 4j). Under scanning electron

microscope, the partially amylose-depleted mutant produced more small-sized starch granules (Fig. 4k and l), and some holes on the starch were also observed (Fig. 4k), which were characteristics of amylose-reduced rice grains[57].

## Comprehensive off-target assessment of LrCas9 in rice

Having demonstrated highly efficient genome editing by LrCas9 in rice, we sought to evaluate its potential off-target effects. First, we assessed LrCas9's tolerance of mismatches in spacers. We systematically introduced 1-bp mismatches at every position of one chosen 20 bp spacer and assessed the ability of genome editing by these

**Fig. 3 | LrCas9 confers efficient genome editing in diverse plant species. a** and **b** The co-target site illustration of LrCas9, LbCas12a, ttLbCas12a and LbCas12a-RRV (**a**), and of SpCas9-NG, SpRY and LrCas9 (**b**). The cut site of LrCas9, SpCas9-NG, and SpRY was labeled with salmon, while the LbCas12a was labeled with cyan. The target sites' detail was listed in Supplementary Data 4. **c** The mutation rate comparison of LbCas12a, ttLbCas12a, LbCas12a-RRV, and LrCas9 at four rice endogenous loci in protoplasts (left). The data were analyzed using an unpaired *t*-test with a two-tailed *P* value. The summarized violin plot of mutation rates by LbCas12a, ttLbCas12a, LbCas12a-RRV, and LrCas9 (right). Solid line, median; dash line, quartiles. Each dot represents a biological replicate. Each target contains three biological replicates. Data are presented as mean values ± s.d. ns, *P* > 0.05; *P* < 0.05; **P* < 0.01; ****P* < 0.0001. **d** The mutation rate comparison of SpCas9-NG, SpRY, and LrCas9 at

seven rice endogenous loci in protoplasts (left). The data were analyzed using two-way ANOVA with Tukey's multiple comparisons test. The summarized violin plot of mutation rate by SpCas9-NG, SpRY, and LrCas9 (right). Solid line, median; dash line, quartiles. Each dot represents a biological replicate. Each target contains three biological replicates. Data are presented as mean values ± s.d. ns, *P* > 0.05; *P* < 0.05; ***P* < 0.001; ****P* < 0.0001. **e**–**g** The mutation rates of LrCas9 at endogenous loci in wheat (**e**), tomato (**f**), and larix protoplasts (**g**). Data were analyzed using a two-tailed unpaired *t*-test. ns, *P* > 0.05; *, *P* < 0.05; ***P* < 0.01; ****P* < 0.001; *****P* < 0.0001. The target sites' details are listed in Supplementary Data 4. Each dot represents a biological replicate. Each target contains two or three independent experiments (*n* = 2 or *n* = 3). Data are presented as mean values ± SD. Source data are provided as a Source Data file.

**Table 1 | The mutation rates of multiplexed genome editing by LrCas9 in rice T₀ lines**

| No. | Site | Target gene | Protospacer + PAM | Tested T₀ lines | Mutated T₀ lines (number; ratio) | Biallelic T₀ lines (number; ratio) | Deletion T₀ lines (number; ratio) |
|---|---|---|---|---|---|---|---|
| pZHZ741 | Os-AG04 | *OsPDS* | AGTCCTGGCAAACAACCTGCAGAAA | 23 | 7; 30.4% | 3; 13.0% | – |
| | Os-CG01 | *OsDEP1* | TCCCGAGCGCGGAGTACGTACGAAA | | 12; 52.2% | 11; 47.8% | – |
| pZHZ742 | Os-TG01 | *OsPDS* | CTAGCCAAGCTATTTCCTGATGAAA | 27 | 23; 85.2% | 22; 81.5% | 4; 14.8% |
| | Os-TG02 | *OsPDS* | TGGCATTTCTACCTTATCGATGAAA | | 19; 70.4% | 8; 29.6% | |

mutated spacers. Interestingly, we found 1-bp mismatches, including those at the 5′ end, largely reduced and destroyed LrCas9's genome editing capability (Fig. 5a). Next, we use GUIDE-seq (Genome-wide, Unbiased Identification of DSBs Enabled by Sequencing), a highly sensitive tool to detect off-target sites[58]. We set up a rice protoplast-based GUIDE-seq system, where NGS was used to map and quantify the integration of double-strand oligonucleotide (dsODN) tags, known as GUIDE, at DSB sites (Fig. 5b). Our data showed that 10 pmol GUIDE concentration was sufficient to yield detectable (~3.5%) GUIDE insertion at the OsGG02 target site (Fig. 5c). Interestingly, we discovered asymmetric GUIDE insertion in this specific case (Fig. 5d). We tested off-target effects of four independent spacers with GUIDE-seq. In all cases, the vast majority of GUIDE insertions (87.8% and 98.8%) were found to be at the target sites (Fig. 5e and Supplementary Data 3). In addition, the captured off-target sites all contained six or more mismatches to the spacers and PAMs (Fig. 5e), making them unlikely to be true off-target sites[9,36]. To further confirm, we directly genotyped T₀ lines edited at Os-AG04 and Os-TG02 sites. The top off-target sites nominated by GUIDE-seq and by CRISPR-GE[59] were checked by Sanger sequencing. At all these sites tested, no off-target mutations were identified (Table 2, Supplementary Table 2, and Supplementary Fig. 14). Hence, these three independent assays collectively demonstrated that LrCas9 is a highly specific nuclease in genome editing.

### Expanding base editing scope with LrCas9 base editors

Base editors such as CBEs and ABEs have greatly enriched plant genome editing toolbox[41]. Base editing scope, which dictates the usefulness of these tools, can be expanded by using Cas9 proteins that recognize different PAMs. Traditionally, this has been achieved in plants with engineered variants of SpCas9, and only occasionally with SaCas9 and its engineered variant[41]. We reasoned that LrCas9 could be a useful Cas protein to develop base editing tools in plants. To make a CBE, LrCas9-D13A nickase was fused with PmCDA1[60,61], as we recently showed that PmCDA1-UGI-based CBE had undetectable off-target effects in rice[62]. Two LrCas9 CBE expression systems were constructed for their applications in rice and wheat, respectively (Fig. 6a). Testing at eight target sites in rice protoplasts showed that up to 35% C-to-T conversion could be achieved (Fig. 6b), whereas very minimal levels of indel byproducts were found (Supplementary Fig. 15). In wheat protoplasts, up to 10% C-to-T editing was detected (Fig. 6c). In both plant species, this LrCas9 CBE favored editing at the 5′ end of the spacers (Fig. 6d and e), which is a characteristic of PmCDA1[60,62]. Further testing

in stable transgenic rice plants at two target sites showed 90.5% and 57.1% editing efficiency, respectively, with biallelic base editing events readily obtained (Fig. 6f and Table 3).

We also generated two versions of LrCas9 ABEs by fusing ecTadA-7.10[63] and ecTadA-8e[9,64] to the N-terminus of LrCas9 nickase (Fig. 6g). Testing at OsCG-01 site in *OsDEP1* showed that LrCas9-based ABE8e based on ecTadA-8e (ABE V2.0), not the earlier ecTadA-7.10 (ABE V1.0), generated A-to-G conversion events in stable T₀ plants (Fig. 6h and Table 3). Thus, efficient ABEs can be engineered when combining LrCas9-D13A and an efficient adenosine deaminase like ecTadA-8e.

### CRISPR-LrCas9-based tools for highly efficient transcriptional repression and activation

The A-rich PAM requirement of CRISPR-LrCas9 also makes it an ideal system for developing transcriptional repression (CRISPRi) and activation (CRISPRa) tools. To engineer a CRISPRi system, we fused to LrCas9 (D13A, H858A), a deactivated LrCas9 (dLrCas9), with repressor domains KRAB[65] and SRDX[66], resulting in dLrCas9-KS (Fig. 7a). To test this CRISPRi system, we targeted three independent genes in rice with sgRNAs targeting a variety of positions, including close to or overlapping the transcription start site (TSS) (e.g., *OsALT2*-gR01 and *OsGW7*-gR01), downstream of the TSS (e.g., *OsGW7*-gR02), right upstream of the TSS (e.g., *OsALT2*-gR02; 273 bp from TSS), or far upstream of the TSS (e.g., *OsWx*-gR03; 1229 bp from TSS) (Fig. 7b). Strikingly, they all showed very potent transcriptional repression in rice protoplasts: the expression levels of all three target genes could be repressed to ~20% or lower (Fig. 7c). To our knowledge, this might represent the most potent CRISPRi system ever developed in plants.

To develop a CRISPRa system from LrCas9, we fused the potent transcriptional activator fusion, TV[12], to the C-terminus of dLrCas9 (Fig. 7d). To test this CRISPRa system, we used six sgRNAs to target four independent genes (1–2 sgRNAs per gene) (Fig. 7e). In all cases, moderate transcriptional activation (2–3-fold) was observed in rice protoplasts (Fig. 7f). Similarly, we also found that when nuclease-active LrCas9-TV (Fig. 7g) was paired with short (14-bp) spacers, genome editing activity was abolished (Fig. 7h). We compared this nuclease-active CRISPRa system (LrCas9-TV) with the dLrCas9 based CRISPRa system (dLrCas9-TV) in rice protoplasts for targeted activation of *OsmiR528* and *OsWx*. Remarkably, 19-fold and 150-fold activation was achieved with the nuclease-active LrCas9-TV system (Fig. 7i), representing 11x and 60x improvement over the dLrCas9-TV system. Hence,

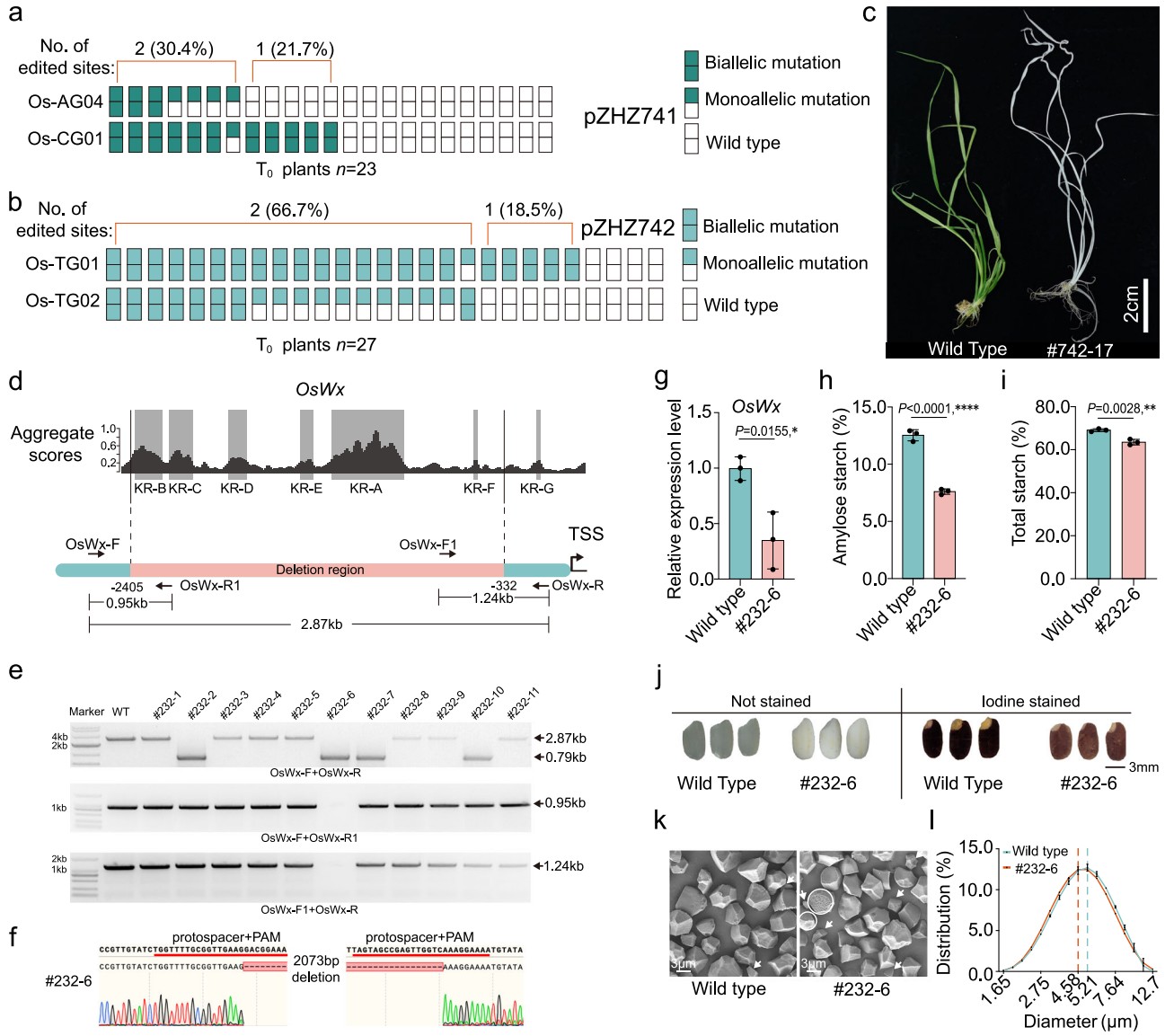

**Fig. 4 | LrCas9-mediated multiplexed coding sequence and promoter editing in stable rice plants. a** and **b** Numbers of edited sites of pZHZ741 (**a**) and pZHZ742 (**b**). **c** The phenotype of a multiplex edited $T_0$ line (#742-17) by LrCas9 at *OsPDS*. **d** The aggregate scores for promoter analysis of *OsWx*. The key regions (KR) were indicated and labeled with gray. The deletion region included 6 KRs, from KR-A to KR-F. The detection primers were shown, and the arrows indicated the direction. The amplicon lengths by different primer sets were shown on the bottom. The prime details are listed in Supplementary Data 5. **e** The PCR-based detection of large promoter deletions in rice $T_0$ lines. The 0.79 kb band indicates deletion events. The missing bands on the middle and bottom gels indicate that line (#232-6) carried homozygous deletion. The PCR was repeated three times independently with the same results. **f** The Sanger sequencing-based confirmation of the #232-6 deletion line. The sgRNA and PAM were underlined with red. **g** The relative expression levels of *OsWx* in wild type and the promoter edited line #232-6. Each dot represents a biological replicate. Each assay contains three independent experiments (*n* = 3). Data are presented as mean values ± SD. **h** and **i** The amylose starch contents (**h**), and total starch contents (**i**) of wild type and #232-6 mature seeds. Each dot represents a biological replicate. Each target contains three biological replicates. Data are presented as mean values ± SD. The data were analyzed using a two-tailed unpaired *t*-test. *$P < 0.05$; **$P < 0.01$; ****$P < 0.0001$. **j** The iodine–starch staining assay of wild type and #232-6 mature seeds. **k** The starch granule in mature seeds under the scanning electron microscope of wild type and #232-6. White arrow indicated the smaller starch granule size under 3 μm. White circle indicated the starch with a small hole due to the low expression level of *OsWx*. The observation was repeated 3 times independently with similar results. **l** The starch granule size distribution of wild-type and #232-6. The dash line in salmon indicated the highest proportion of starch granule diameter of #232-6 at 4.58 μm, while the dashed line in cyan indicated the highest proportion of starch granule diameter of the wild type at 5.21 μm. Each dot represents a biological replicate. Each measurement contains three independent experiments (*n* = 3). Data are presented as mean values ± SD. Source data are provided as a Source Data file.

we developed a highly potent CRISPRa system based on nuclease-active LrCas9 and truncated spacers.

## Discussion

CRISPR-Cas9 and Cas12a are the two main nuclease systems used in plant genome editing. Compared to Cas9, Cas12a lacks nickase versions and its crRNA restricts further modifications. For these reasons, CRISPR-Cas9 is a more versatile genome engineering system that goes far beyond targeted mutagenesis. Currently, base editors, prime editors, CRISPRi, and CRISPRa used in plants are predominantly based on CRISPR-Cas9[35,41]. SpCas9 is the most widely used Cas9, partly due to its simple 5′-NGG-3′ PAM and high nuclease activity. While this PAM requirement is sufficient for targeted mutagenesis to knock out genes, it often falls short of directing precise modifications at well-defined sites by base editors and prime editors. Consequently, many SpCas9 variants, such as SpRY, Cas9-NG, and SpG, have been engineered to

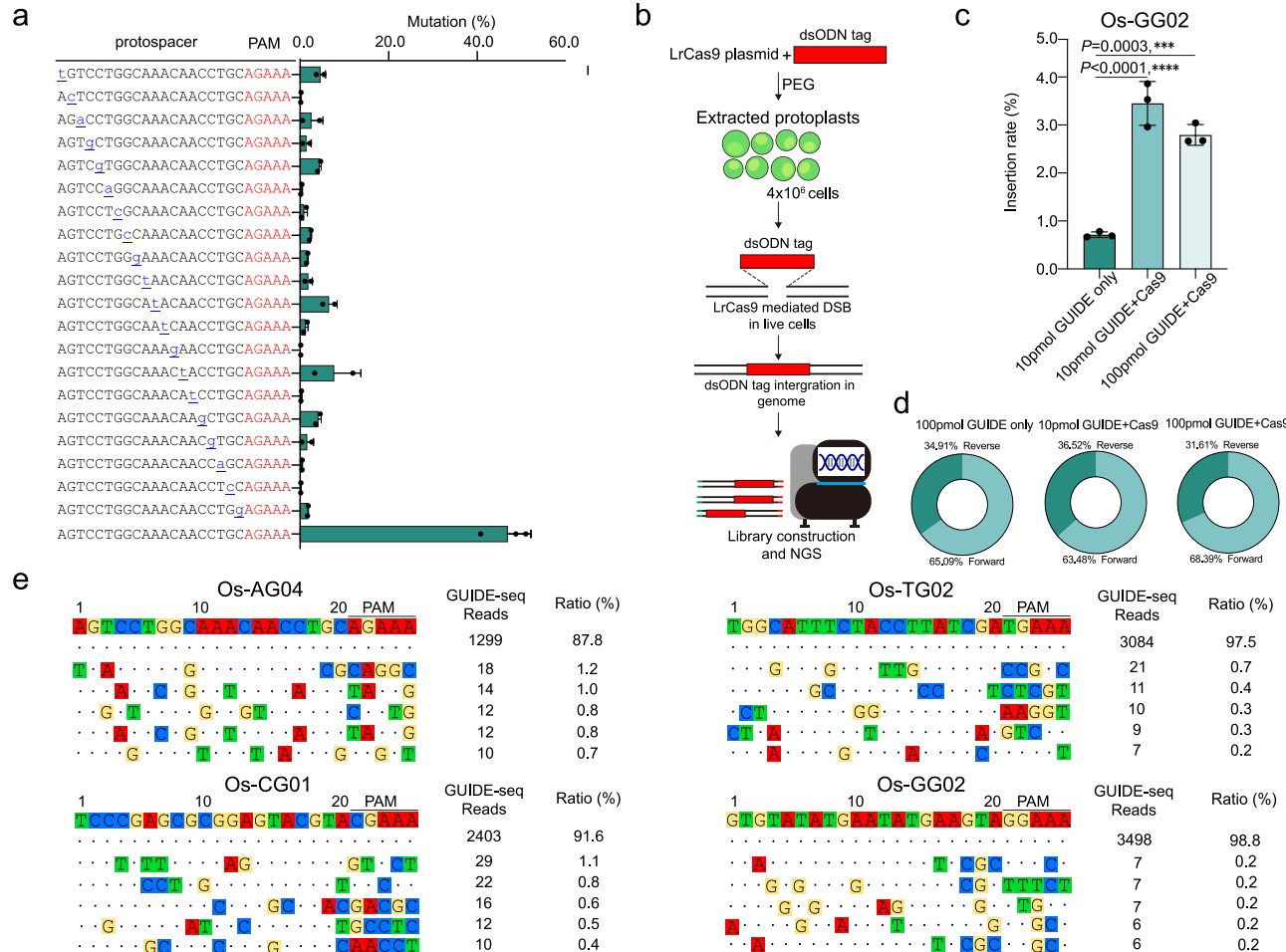

**Fig. 5 | Off-target analyses of the LrCas9 nuclease system in rice. a** The Off-target analysis of LrCas9 at an NGAAA PAM site with mismatched protospacers (spacers) through amplicon deep-sequencing. The single base mismatch was underlined and labeled with blue. The correct spacer along with all 20 single-nucleotide mismatched spacers were compared. Each dot represents a biological replicate. Each spacer contains two or three independent experiments (*n* = 2 or *n* = 3). Data are presented as mean values ± SD. **b** The diagram of GUIDE-seq in this study. See details in the "Methods" section. **c** The dsODN tag (GUIDE) insertion rates at the Os-GG02 site were quantified by deep sequencing. Each dot represents a biological replicate. Each target contains three biological replicates. Data are presented as mean values ± s.d. Data were analyzed by one-way ANOVA with Dunnett's multiple comparisons tests. ***$P < 0.001$; ****$P < 0.0001$. **d** The insertion direction of GUIDE that integrated at the Os-GG02 site in the rice genome. The insertion rates were quantified by deep sequencing. **e** The top five off-target sites identified by GUIDE-seq at Os-AG04, Os-CG01, Os-TG02, and Os-GG02. The dot indicated the same nucleotide residues with the on-target site. The ratios were calculated on the right. Total off-target sites were listed in Supplementary Data 3. Source data are provided as a Source Data file.

target alternative PAMs[7,29,31,35]. However, these SpCas9 variants appear to have reduced editing activity and hence are not as reliable as SpCas9. Limited number of Cas9 orthologs have been used for plant genome editing, such as SaCas9 (recognizing 5′-NNGRRT-3′ PAMs) and ScCas9 (recognizing a 5′-NNG′ PAM)[35,41]. Thus, it is desirable to develop more Cas9 orthologs for plant genome engineering with alternative PAM requirements.

Previously, nearly all these CRISPR systems were first developed for genome editing in human cells and later adopted for use in plants. However, genome editing in plants differs from that in mammalian cells in multiple aspects. First, many of the Cas proteins have optimal working temperatures higher than room temperature (like 25 °C). Hence, a Cas nuclease that works well in human cells may not always work in plants. For example, SpCas9 possesses much higher activity with high-temperature treatment regimes in plants[67]. In another example, AsCas12a works as efficiently as LbCas12a for genome editing in human cells[68], yet it barely works in many plant species when genome editing experiments were carried out at lower temperatures[35]. Second, genome editing in humans has a great focus on the correction of single nucleotide polymorphisms (SNPs) related

to diseases. In contrast, genome editing in plants is often presented as a powerful breeding tool for crop improvement. One promising application for crop improvement is to introduce quantitative traits either through Cas9[54] or Cas12a[39]. Considering the unique needs of plant genome engineering, it will be useful to develop a specific CRISPR-Cas9 system that can target A/T-rich promoter regions and can work well at a relatively low temperature. Third, SpCas9, despite its great success, is sourced from a human pathogen. It may taint public perception when SpCas9-engineered foods enter global markets.

It is due to all these aspects and considerations that we strive to conduct a de novo discovery approach to identify an effective CRISPR-Cas9 system that can enrich the plant genome engineering toolbox. Our bioinformatic and empirical analyses identified Cas from *Lactobacillus rhamnosus* GG (LrCas9), which recognizes a unique 5′-NGAAA-3′ PAM that is distinct from all known Cas9 proteins used in genome editing. We showed that LrCas9 is relatively tolerant to lower temperatures and confers efficient genome editing in a variety of plant species, including rice, wheat, tomato, and larix. Remarkably, when targeting the same sequences, we found that LrCas9 conferred

**Table 2 | Off-target analysis in rice T0 lines of Os-AGO4 and Os-TGO2 by GUIDE-seq and CRISPR-GE prediction**

| Index | Chromosome | Position | Method | Sequence | PAM | Gene_ID | Region | Edited |
|---|---|---|---|---|---|---|---|---|
| On-target | Chr03 | 4410363 | – | AGTCCTGGCAAACAACCTGC | AGAAA | LOC_Os03g08570 (Os-AGO4) | Intron | Yes |
| Off-target1 | Chr08 | 17027595 | GUIDE-seq | AGTACTCGGAATCAACATGC | TAAAG | LOC_Os08g27940 | Intron | No |
| Off-target2 | Chr03 | 22118232 | GUIDE-seq | AGGCTTGGCGAAGTACCTGC | CGATG | – | Intergenic | No |
| Off-target3 | Chr05 | 15939244 | GUIDE-seq | AGTACTCGGAATCAACATGC | TAAAG | – | Intergenic | No |
| Off-target4 | Chr02 | 32932893 | GUIDE-seq | AGTCGTGGCTAACTAACTGG | AGGAT | – | Intergenic | No |
| Off-target5 | Chr10 | 13417495 | CRISPR-GE | AGTCATGGCAATCGACCAGC | AGAAA | LOC_Os10g25890 | Intron | No |
| Off-target6 | Chr04 | 7972534 | CRISPR-GE | AGCACTGGAAAACACCCTGC | AGAAA | – | Intergenic | No |
| Off-target7 | Chr06 | 9197794 | CRISPR-GE | ATTCGTGACAAAAAAACTGC | AGAAA | – | Intergenic | No |
| Off-target8 | Chr05 | 6095578 | CRISPR-GE | AGTTCCGTCAAACAAACTTC | AGAAA | – | Intergenic | No |
| On-target | Chr03 | 4411032 | – | TGGCATTTCTACCTTATCGA | TGAAA | LOC_Os03g08570 (Os-TGO2) | Intron | Yes |
| Off-target1 | Chr05 | 29520552 | GUIDE-seq | TGGGATTGCTATTGTATCGA | CCGAC | LOC_Os05g51490 | Intron | No |
| Off-target2 | Chr05 | 19638874 | GUIDE-seq | TGGCATGCCTACCTCCTCGT | CTCGT | LOC_Os05g33420 | CDS | No |
| Off-target3 | Chr08 | 589207 | GUIDE-seq | TCTCATTTCGGCCTTATCGA | AAGGT | – | Intergenic | No |
| Off-target4 | Chr08 | 24642717 | GUIDE-seq | CTGAATTTCTTCCTTATCAA | GTCAA | LOC_Os08g38980 | CDS | No |
| Off-target5 | Chr03 | 27025699 | GUIDE-seq | TGGAATTTGTACCATATCCA | TGAAT | LOC_Os03g47693 | Intron | No |
| Off-target6 | Chr03 | 24493324 | CRISPR-GE | TGGCATTTATAGATTATTGA | AGAAA | – | Intergenic | No |
| Off-target7 | Chr10 | 11655500 | CRISPR-GE | TGAGAATCCTACCTTATCAA | AGAAA | LOC_Os10g22484 | CDS | No |
| Off-target8 | Chr05 | 1265737 | CRISPR-GE | TTGAATTCCCACGTTATCGA | AGAAA | – | Intergenic | No |

higher editing efficiency than LbCas12a, SpCas9-NG, and SpRY, all of which are routinely used in genome editing.

Given the relatively complex PAM, one may wonder about the usefulness of LrCas9 as a genome editing tool. We conducted an *in-silico* analysis to compare LrCas9 and SpCas9 on targetability in six major crops (rice, maize, tomato, soybean, tobacco, and wheat). We found that LrCas9 can target as many genes as SpCas9, and it was when the number of target sites per gene increased, that we started to see differentiation among these two Cas9 systems (Supplementary Fig. 16). Since the targetable sites by LrCas9 and SpCas9 barely overlap, it is beneficial to have LrCas9 as a useful addition to the CRISPR-Cas9 toolbox. It is essential to enrich the base editing toolbox with base editors that can recognize alternative PAMs. In this regard, we successfully demonstrated LrCas9-derived CBE and ABE in rice and wheat. Prime editing[69] is useful for molecular breeding in plants. We made a LrCas9 prime editor 2 (PE2) by linking LrCas9 nickase (H858A) with an M-MLV reverse transcriptase and using an epegRNA scaffold[70] (Supplementary Fig. 17a). We successfully detected the desired C to A conversion with 0.1% editing efficiency in rice protoplasts (Supplementary Fig. 17b and c). The low prime editing efficiency could be due to the relatively long LrCas9's sgRNA scaffold of 139nt, as compared to SpCas9's sgRNA2.1 scaffold of 92nt. The longer scaffold may form a more complex secondary structure that may strongly influence the prime editing efficiency[71]. Further work to enhance the LrCas9-based prime editor is warranted.

LrCas9 recognizes an A/T-rich PAM, making it an alternative system for targeting promoter sequences in plants. Indeed, after developing a multiplexed CRISPR-LrCas9 system, we used it to create a large chromosomal deletion in the promoter of *OsWx*, which significantly reduced the *OsWx* expression and generated rice grains with less amylose content. We further explored this CRISPR-LrCas9 platform to develop efficient CRISPRi and CRISPRa systems for efficient transcriptional repression and activation, respectively. We found CRISPRi based on dLrCas9-KS is highly efficient, generating ~80% or more reduction in transcripts for target genes based on assays in rice protoplasts. Interestingly, our CRISPRa system based on nuclease-active LrCas9-TV and truncated spacers of 14 bp showed much higher transcriptional activation potency than the dLrCas9-TV system. This puts LrCas9 in a great position to be further developed into a CRISPR-Cas-based regulation system that would enable simultaneous genome

editing and gene activation[7]. Interestingly, we also found that LrCas9 displayed variable genome editing efficacies at non-canonical and shorter 5'-NGAA-3' PAMs in rice cells (Supplementary Fig. 18). Hence, as with SpCas9, we believe LrCas9 can be further improved with RNA engineering and protein engineering to recognize more diverse PAMs for promoter engineering and gene regulation.

Since CRISPR-LrCas9 is a currently developed genome engineering system, it is critical to investigate its potential off-target effects[9,36]. We employed three independent approaches to assess this. First, we introduced mismatched 1-bp mutations at every position of the 20-bp spacer. To our surprise, LrCas9 is less tolerant to 1-bp mutations, and this result is even better than the high-fidelity xCas9 evaluated in a similar assay[29]. This high specificity was further validated with the unbiased GUIDE-seq analysis, where not a single convincing off-target site was identified for all four independent spacers. Finally, we evaluated LrCas9-edited rice plants and could not find any off-target mutations on potential off-target sites nominated by either GUIDE-seq or CRISPR-GE. Hence, CRISPR-LrCas9 is a highly precise genome engineering system. Its unique PAM and spacer's less tolerance to mismatch mutations could collectively contribute to its high targeting specificity.

The widely used SpCas9 is sourced from *Streptococcus pyogenes*[1], which is a major human pathogen. Indeed, humans have developed adaptive immunity against SpCas9 protein[72]. By contrast, *Lactobacillus rhamnosus GG*, where our LrCas9 is sourced from, is a common probiotic that is widely consumed by humans due to its health benefits[73]. CRISPR-Cas systems hold great promise to boost crop breeding and enhance nutrition in our food. This bright future however will not be easily realized without the de-regulation of these genome engineering tools and the resulting products. We hope the use of this CRISPR-LrCas9 genome engineering system from a beneficial probiotic bacterial strain may help promote a favorable public perception of genome-edited crops.

## Methods
### Data mining
The Hidden Markov Models (HMMs)[74] were used to analyze the CRISPR-Cas sequences. The models were trained and constructed using the seed sequences by TIGRFAMs[75] (total number of 101) and Pfam[76] (total number of 38) database of CRISPR-Cas for known Cas proteins. By using HMMs with the default parameters, the proteomes

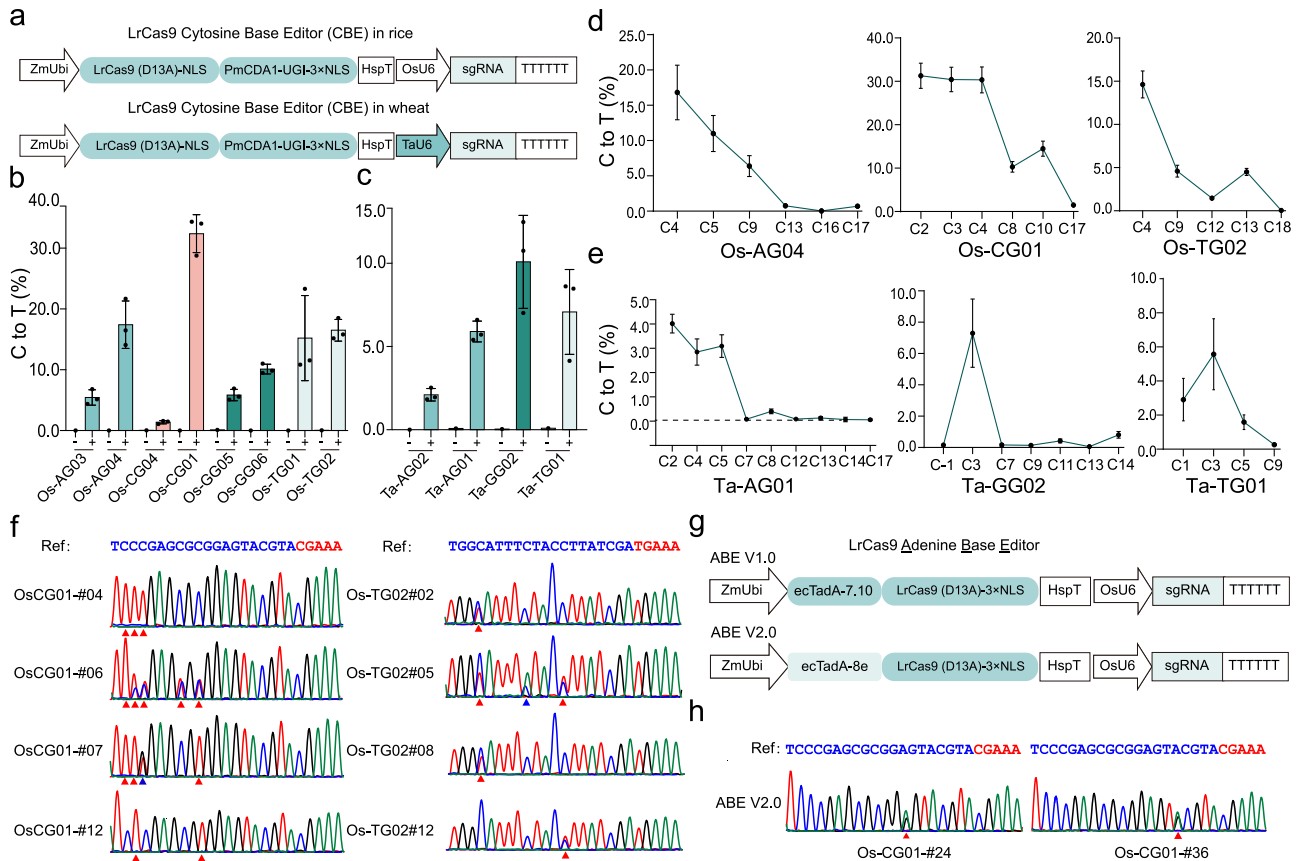

**Fig. 6 | Assessment of LrCas9-based cytosine base editors and adenine base editors in plants. a** Diagrams of LrCas9-based CBE in rice (top) and wheat (bottom). **b** and **c** C-to-T editing rates by LrCas9 CBE at endogenous loci in protoplasts of rice (**b**) and wheat (**c**). Each dot represents a biological replicate. Each target contains three biological replicates. Data are presented as mean values ± s.d. **d** and **e** The C-to-T editing rates in different C's positions in rice (**d**) and wheat (**e**). Each target contains three independent experiments ($n = 3$). Data are presented as mean values ± SD. **f** Sanger sequencing results of select rice $T_0$ lines with targeted base editing. The red triangle indicated C-to-T conversion, and the blue triangle indicated C-to-G (OsCG01-#07) or C-to-A (OsTG02-#05) conversion. **g** The diagrams of LrCas9-based ABE V1.0 and V2.0 for expression in rice. **h** Sanger sequencing results of two A-to-G base edited rice $T_0$ lines by ABE V2.0. Red triangle indicates A-to-G conversion. Source data are provided as a Source Data file.

**Table 3 | Base editing frequency by LrCas9-based CBE and ABE in rice $T_0$ lines**

| Reagent | Target gene | Site | protospacer + PAM | Tested $T_0$ lines | Mutated $T_0$ lines (number; ratio) | Base editing $T_0$ lines (number; ratio) |
|---|---|---|---|---|---|---|
| CBE | *OsDEP1* | Os-CG01 | TCCCGAGCGCGGAGTACGTACGAAA | 21 | 19; 90.5% | 8; 38.0% |
| | *OsPDS* | Os-TG02 | TGGCATTTCTACCTTATCGATGAAA | 21 | 12; 57.1% | 4; 19.1% |
| ABE V1.0 | *OsDEP1* | Os-CG01 | TCCCGAGCGCGGAGTACGTACGAAA | 23 | 0; 0.0% | 0; 0.0% |
| ABE V2.0 | | | | 14 | 2; 14.3% | 2; 14.3% |

of each species obtained from GenBank (https://ftp.ncbi.nih.gov/genomes/genbank/bacteria and https://ftp.ncbi.nih.gov/genomes/genbank/archaea/) were analyzed, which resulted in the identification of 257,745 putative Cas proteins, and 30,495 CRISPR clusters. Further, CRISPRFinder[77] was used for a genome-wide screen to identify potential CRISPR arrays adjacent to these clusters[78]. The PAM features were obtained by searching the anti-spacers in virus/phage genomes using BLAST tools for each putative CRISPR-Cas system and the PAM sequences were visualized using the WebLogo tools[45]. The species were prioritized with a high degree of certainty regarding the PAM sequences and very similar species were excluded from our analysis. After applying these criteria, 42 putative CRISPR-Cas systems were selected as candidates. Jalview[79] was used to analyze the CRISPR RNA from 42 putative CRISPR-Cas systems and the feature sequence was drawn by WebLogo. The resulting phylogenetic tree for the 42 Cas9 proteins was displayed and annotated using the Interactive Tree Of

Life (iTOL) tool[80]. The modeling of five Cas9 protein structures (CAL43592.1, BAI42646.1, OFH73969.1, EPW83356.1, AAK33936.1) was performed with SWISS-MODEL[46].

**Vector construction**

The vectors were constructed based on the backbone of pGEL062 (Addgene#124894). The DNA sequences of chosen Cas9 genes were rice codon optimized and synthesized. The dCas9-TV[12] was requested from Jianfeng Li's lab. All the DNA fragments were constructed into linearized pGEL062 backbone by Gibson Assembly. For T-DNA vector construction, the spacer oligos were synthesized by Sangon Biotech (Shanghai, China) and annealed for standard Golden Gate reaction[81] followed by DH5a *E.coli* transformation. The fragment for multiplex editing, flanked with *Bsa*I restriction enzyme sites, was synthesized, and the final T-DNA vectors were constructed with Golden Gate reaction[82]. For the editing activity test in rice protoplasts of four Cas9

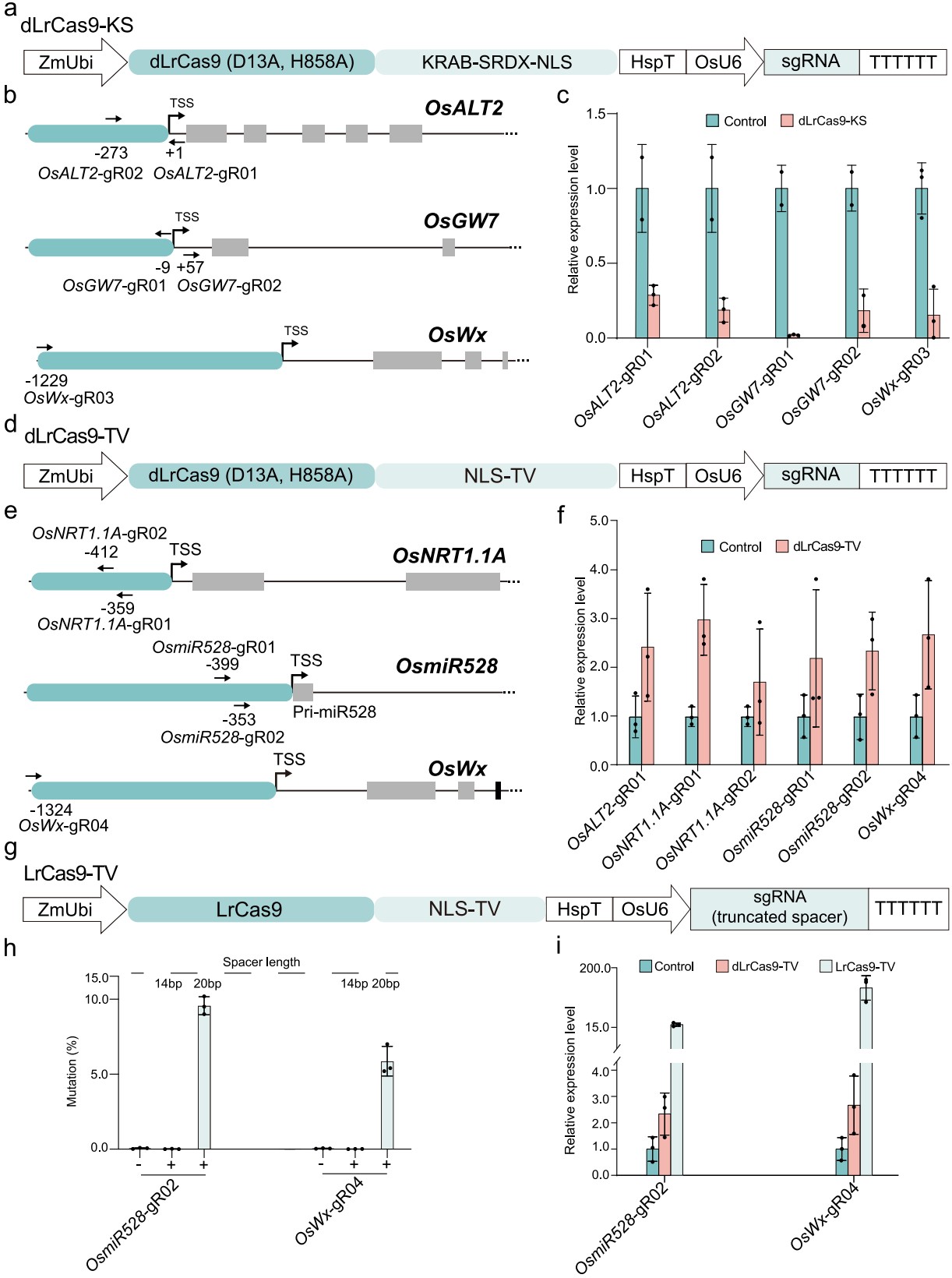

candidates, the crRNAs and tracrRNAs were driven by an *OsU6* promoter and further processed by tRNA, while the Cas9 genes were driven by the Ubiquitin promoter of *Zea mays*[83]. For PAM depletion assay in *E. coli*, the crRNAs and tracrRNAs were driven by the J23119 promoter separately, and the LrCas9 genes were driven by the J23100 promoter. All the vectors were confirmed by Sanger sequencing.

**PAM depletion assay**

The LrCas9 expression vector and empty control vector (no LrCas9) were first transformed into *E. coli* DH5a to make the competent cell with Kanamycin selection. Then, 10 ng target plasmid that contained the crRNA and tracrRNA expression cassette and the target sequence were transformed into the competent cell by the heat shock method.

**Fig. 7 | LrCas9-based transcriptional repression and activation systems in plants. a** Diagrams of a dLrCas9-KS CRISPRi system for expression in rice. The details for T-DNA expression cassettes are shown in Supplementary Fig. 17. **b** The sgRNAs' binding sites for gene repression. The sgRNA's directions were indicated by arrows, and the distance to TSS was shown for each sgRNA. The promoter areas were filled in cyan and the exons were filled with gray. **c** The gene relative expression levels of dLrCas9-KS compared with the control in rice protoplasts quantified by real-time qPCR. Each dot represents a biological replicate. Each target contains two or three independent experiments ($n = 2$ or $n = 3$). Data are presented as mean values ± SD. **d** Diagrams of a dLrCas9-TV CRISPRa system for expression in rice. **e** The sgRNAs' binding sites for gene activation. The sgRNA's directions were shown by arrows, and the distance to TSS was shown for each sgRNA. **f** The gene relative expression levels of dLrCas9-TV compared with the control in rice protoplasts quantified by real-time qPCR. Each dot represents a biological replicate. Each target contains three biological replicates. Data are presented as mean values ± s.d. **g** Diagrams of the nuclease-active LrCas9-TV CRISPRa system with truncated protospacers (spacers). **h** The mutation rates with 14 bp spacer and 20 bp spacer at *OsmiR528* and *OsWx*. Each dot represents a biological replicate. Each design contains three independent experiments ($n = 3$). Data are presented as mean values ± SD. **i** The gene relative expression levels by dLrCas9-TV, nuclease-active LrCas9-TV compared with the control in rice protoplasts quantified by real-time qPCR. Each dot represents a biological replicate. Each target contains three biological replicates. Data are presented as mean values ± s.d. Source data are provided as a Source Data file.

After 1 h incubation at 37 °C, the mixture was plated on the Kanamycin and Ampicillin double selection LB solid medium. The plates were kept at 37 °C. After 16 h of growth, the colony numbers were counted by ImageJ (https://imagej.net/) software.

## Protoplast transformation
In this study, we utilized various plant species, including the rice variety Nipponbare, wheat variety Chinese Spring, tomato variety Alisa Craig, and Larix species. The protocols for protoplast transformation were in accordance with previously published methods[31,84]. For rice and wheat, the process involved germinating sterilized seeds on 1/2 MS solid medium for 10 days in a dark chamber at 28 °C. Subsequently, healthy leaves were finely sliced into strips measuring 0.5–1.0 mm, followed by immersion in an enzyme solution and vacuum infiltration for 30 min. In the case of tomatoes, seeds were cultivated in the soil at 26 °C under an 8-h dark and 16-h light cycle for a duration of 30 days. Healthy leaves were then meticulously cut into strips of 0.5–1.0 mm, immersed in an enzyme solution, and incubated at 40 rpm for 8 h at 28 °C in darkness. To facilitate Larix protoplast transformation, Larix seeds were subjected to 14 days of darkness at 26 °C to induce callus formation. Larix callus tissues were subsequently transferred into the enzyme solution and hydrolyzed for 6 h. The resulting enzyme/protoplast solution was filtered through a 70 μm nylon mesh. During the protoplast transformation process, 30 μg of plasmid DNA in 30 μL was employed to transform 200 μL of protoplasts ($2 \times 10^5$ protoplasts). These were gently mixed with 230 μL of a 40% PEG–CaCl₂ transformation buffer. After incubation for 30 min in darkness, the reactions were quenched by adding 900 μL of W5 washing buffer. The protoplasts were subsequently centrifuged and transferred into a 12-well culture plate, followed by incubation at 32 or 28 °C in darkness for a period of 2 days. The protoplast transformation efficiencies were ~90% for rice and tomato, and ~40% for wheat and larix. After 48 h post-transformation, the protoplasts were harvested for DNA extraction by the CTAB method[85].

## Rice stable transformation
The stable transformation of rice was conducted following previously published protocols[86,87]. In brief, dehulled rice seeds were subjected to sterilization and subsequently cultured on an N6D solid medium. Rice calli, which had been pre-cultured, were transformed through inoculation with Agrobacterium EHA105 carrying the recombinant vector. Following a co-incubation period of 3 days between the rice calli and Agrobacterium, the calli were rinsed with sterilized water and transferred to N6-S medium for a 2-week selection. The newly grown calli were then transferred to RE-III medium for a 2-week cultivation. Resistant calli were subsequently moved to fresh RE-III medium every 2 weeks until regenerated plants were successfully obtained.

## RNA sequencing
The CRISPR array and tracrRNA region of *Lactobacillus rhamnosus GG* was PCR amplified by *ApexHF* HS DNA Polymerase (Accurate Biology,

China) and TA-cloned into pEASY-T1 cloning kit (TransGen, China), following the manufacturer's instruction. The colonies confirmed by Sanger sequencing were cultured in the flask for 16 h. The cultured bacteria were collected by centrifuge and remove the supernatant. After fast freezing by liquid nitrogen, the samples were sent to Novogene (China, Tianjin) for RNA sequencing using the Novaseq6000 platform. The histogram was exported by the Integrative Genomics Viewer (IGV)[88] software.

## Mutagenesis analysis
The mutation frequency in protoplasts was analyzed by amplicon deep-sequencing[61]. The amplicons of the editing regions were amplified by *ApexHF* HS DNA Polymerase (Accurate Biology, China), and the barcodes were added through PCR primers. Amplicons were sent to Novogene (China, Tianjin) for deep-sequencing by the Novaseq6000 platform which produced 150 bp paired-end reads. The editing frequency and profile were analyzed by CRISPRMatch software[89]. The mutation frequencies in wheat and larix were normalized to the protoplast transformation efficiencies in both species. For stable rice $T_0$ lines, the editing outcomes were identified by single-strand conformation polymorphism assay[82] and direct PCR product Sanger sequencing provided by Sangon Biotech (Shanghai, China) and analyzed by CRISPR-GE DSDecodeM software[59].

## GUIDE-seq
The GUIDE-seq experiment was conducted by following the previous research protocol[58] with slight modifications. Briefly, the different concentration GUIDE and 30 ug LrCas9 plasmids were co-transformed into rice protoplasts. The DNA was extracted after 48 h incubation at 32 °C. The libraries were constructed by using TruePrep DNA Library Prep Kit (Vazyme, China) following the manufacturer's instructions. Constructed libraries were sent to Novogene (China, Tianjin) for deep sequencing. The GUIDE-seq results were obtained by analyzing the raw sequencing reads using Python scripts from Tsai's Lab (https://github.com/tsailabSJ/guideseq).

## Real-time quantitative PCR
The total RNA was extracted by using the *SteadyPure* plant RNA isolation kit (Accurate Biology, China), and the first strand cDNA was synthesized by using the HiScript III 1st Strand cDNA Synthesis Kit (Vazyme, China). The real-time qPCR was performed using ChamQ Universal SYBR qPCR Master Mix (Vazyme, China), and the relative expression levels were calculated by the $2^{-\Delta\Delta CT}$ method.

## Data analysis
The data were analyzed using GraphPad Prism 8.0 software, and the figures were further processed using Adobe Photoshop and Adobe Illustrator software.

## Reporting summary
Further information on research design is available in the Nature Portfolio Reporting Summary linked to this article.

## Data availability

The vectors are available from Addgene, with the Addgene numbers listed in Supplementary Fig 19. The high-throughput sequencing data sets generated from this study are available at NCBI Sequence Read Archive under Bioproject PRJNA1017971 or China National GeneBank DataBase under CGNB Project CNP0004270. Source data are provided with this paper.

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

## Acknowledgements

This research was supported by the National Natural Science Foundation of China (award nos. 32270433, 32072045, and 31960423) to Y.Z. and X.Z., the China Postdoctoral Science Foundation (award no. 2022M720663) and the Sichuan Postdoctoral Special Foundation (award no. TB2022015) to Z.Z., the Technology Innovation and Application Development Program of Chongqing (award no. CSTC2021JSCX-CYLHX0001) to Y.Z., the Incubation Program for Innovative Science and Technology in UESTC (award no. Y03023206100209) to J.H., the Open Foundation of Jiangsu Key Laboratory of Crop Genetics and Physiology (award no. YCSL202009) to Y.Z and T.Z. It is also supported by the NSF Plant Genome Research Program (IOS-2029889 and IOS-2132693) and USDA-NIFA (2021-67013-34554) to Y.Q.

## Author contributions

Y.Z. proposed the project and designed the experiments. T.T., Z.Z., J.H. and Y.Z. predicted and analyzed the CRISPR loci. Z.Z. and S.X. constructed all the plasmids. Z.Z. and Y.H. performed the rice and wheat protoplast transformation. L.Y. and Z.Z. performed tomato protoplast transformation. L.H. performed the larix protoplast transformation.

G.L. and T.Z. analyzed the mutation frequencies in protoplasts, the GUIDE-seq results, and the targeting scope in crop genomes. Z.Z., S.X., Y.H., S.L., T.F., X.Z. performed the rice stable transformation. Z.Z. and S.X. analyzed the genotypes of rice T0 lines. Y.Z., Y.Q., Z.Z. and J.H. analyzed the data and wrote the manuscript with input from other authors. All authors read and approved the final version of the manuscript.

## Competing interests

The authors declare no competing interests.
