## [Peer Review File · Nature Communications]

A probiotic sourced CRISPR-Cas9 system confers efficient plant genome engineeringEditorial Note: This manuscript has been previously reviewed at another journal that is not operating a transparent peer review scheme. This document only contains reviewer comments and rebuttal letters for versions considered at Nature Communications.

Reviewers' Comments:

Reviewer #2:

Remarks to the Author:

The authors have responded to all of the comments, and most of my concerns have been addressed. However, there are a few minor issues that should be taken into consideration.

1. Needs to clearly label in the Figure Legend whether the editing efficiency shown in figures (e.g., Fig. 3e-3g) is for protoplasts or T0 transgenic plants.
2. The article does not provide any information on the vector (Vector No. Pzhz741/742) or its construction method, despite mentioning it in Figure 4a.
3. Some minor errors: line 497; Figure 4c, T0 plants.

Reviewer #3:

Remarks to the Author:

I read the revised version of manuscript of Zhong et al as well as their rebuttal letter with great interest. The authors took up most suggestions of the 4 reviewers and they replied to their concerns raised in a constructive way. They incorporated a row of new data including prime editing experiments as well as a comparison of LrCas9 to mutated LbCas12a variants. All this shows that although in one or the other aspect the "classical" enzymes might perform better, LrCas9 might indeed be developed to an interesting alternative to SpCas9 and LbCas12a. Thus, the revised manuscript well deserves a wide audience. It will be especially interesting to see whether its editing activity can be tuned by mutagenesis and how efficient repressors based on the enzyme might become.

Reviewer #4:

Remarks to the Author:

The revised manuscript submitted by Zhong et al. describes the identification and application of a new CRISPR-Cas9 system, LrCas9, in crop plants. The research is well designed, and the conclusions are not controversial. However, I am concerned about the novelty of the research. Although this LrCas9 is a new Cas9 system, mining of novel CRISPR-Cas systems has been widely carried out and published. In addition, base editor, CRISPRi, and CRISPRa systems have also been reported.

LrCas9 recognizes NGAAA PAM sequence. However, its mutational efficiency for CGAAA PAMs seems to be much lower than others (Fig. 2h, i, Sup Fig. 7). A statistical analysis of Figure 2i must be performed.

If the insertions are the majority of mutations caused by LrCas9, the results should be shown in the main figure instead of deletions (Fig. 2j, k). The text should also be modified.

Unify the presentation, e.g. sgRNA+PAM (Fig. 4a) and spacer PAM (Fig. 5a), etc.

Some of the sgRNA sequences in the figure are not identical, for example, OsTG01 (Fig. 4a, Fig. 5e, f) and OsTG02 (Fig. 6f, g). I would strongly recommend the authors to really carefully check entire

manuscript and all the data.

The results showed that the LrCas9-mediated CRISPRi suppressed the expression of target genes by ~20% or less (Fig. 7c). On the other hand, mutation efficiency in protoplasts was 20-50% (Fig. 2, 3). I am wondering why the LrCas9-mediated CRISPRi system is so efficient in protoplasts. As authors mentioned, rice protoplast transformation efficiency was ~90%. Does this mean that the suppression efficiency of the CRISPRi system via LrCas9 is almost 100%?

REVIEWERS' COMMENTS

Reviewer #2 (Remarks to the Author):

The authors have responded to all of the comments, and most of my concerns have been addressed. However, there are a few minor issues that should be taken into consideration.

Response: We are very grateful to the Reviewer #2's kind help to improve our manuscript, and we also thank you for the professional comments on this work. Our response are as follows.

1. Needs to clearly label in the Figure Legend whether the editing efficiency shown in figures (e.g., Fig. 3e-3g) is for protoplasts or T0 transgenic plants.

Response: Thanks for your kind suggestion. We added extra explanation in the Figure Legend to clearly described that the editing efficiency result are in protoplasts of the three plant species .

2. The article does not provide any information on the vector (Vector No. Pzhz741/742) or its construction method, despite mentioning it in Figure 4a.

Response: We are sorry for the missed information. The pZH741 and pZH742 vectors were constructed by Golden Gate reaction. We added the details in the Method part of the manuscript.

3. Some minor errors: line 497; Figure 4c, T0 plants.

Response: Thanks for your careful reading. We fixed it in the our revision.

Reviewer #3 (Remarks to the Author):

I read the revised version of manuscript of Zhong et al as well as their rebuttal letter with great interest. The authors took up most suggestions of the 4 reviewers and they replied to their concerns raised in a constructive way. They incorporated a row of new data including prime editing experiments as well as a comparison of LrCas9 to mutated LbCas12a variants. All this shows that although in one or the other aspect the “classical” enzymes might perform better, LrCas9 might indeed be developed to an interesting alternative to SpCas9 and LbCas12a. Thus, the revised manuscript well deserves a wide audience. It will be especially interesting to see whether its editing activity can be tuned by mutagenesis and how efficient repressors based on the enzyme might become.

Response: We are very grateful for Reviewer #3’s helpful, pertinent, and professional advice, and thanks for your appreciation on this work. We agree that our work had been improved significantly under your suggestion and we believe the LrCas9 will become another important CRISPR-Cas9 system for genome engineering in plants and beyond. Thank you!

Reviewer #4 (Remarks to the Author):

The revised manuscript submitted by Zhong et al. describes the identification and application of a new CRISPR-Cas9 system, LrCas9, in crop plants. The research is well designed, and the conclusions are not controversial. However, I am concerned about the novelty of the research. Although this LrCas9 is a new Cas9 system, mining of novel CRISPR-Cas systems has been widely carried out and published. In addition, base editor, CRISPRi, and CRISPRa systems have also been reported.

Response: We appreciate Reviewer #4's time and effort on commenting our work. Just as you said, that there is no controversial conduction in this study. Like other reviewers, we believe our work has sufficient novelty that has been well spelled out and demonstrated in this comprehensive study. Like we wrote on the previous response, based on the original LrCas9 system, we achieved versatile functional application in diverse plant species and prove its suitability in plants genome editing. Meanwhile, the original LrCas9 could edit in a higher efficiency compare with current wild type and evolved CRISPR-Cas systems, and the LrCas9 system also proved to have high specificity. Of note, these LrCas9-based tools meant for engineering plants, for the first time, are coming from a probiotic, not from human pathogens. These features make LrCas9 more attractive in future application for crop improvement and food innovation. Yes, there were many new Cas9 mining work published these years. However, we consider our LrCas9 work is one of the most comprehensive studies and we provide a new system unlike anything published before. For all these reasons, we believe this revised manuscript is worth to be considered for publishing at Nature Communications.

LrCas9 recognizes NGAAA PAM sequence. However, its mutational efficiency for CGAAA PAMs seems to be much lower than others (Fig. 2h, i, Sup Fig. 7). A statistical analysis of Figure 2i must be performed.

Response: Thank you for your suggestion. To initiate the assessment of editing activity, our goal was to demonstrate the broad compatibility of LrCas9 with all NGAAA PAM sequences. However, even though the data show variable activity at different PAM loci, our findings still indicate that LrCas9 exhibited consistent mean editing efficiency across the four various NGAAA PAMs without significant difference. To clearly convey this message, we followed your advice to add the statistical analysis label in Figure 2i, which revealed that there were no significant difference in editing efficiency among the four PAM groups, as confirmed by a one-way ANOVA test with the Tukey method.

If the insertions are the majority of mutations caused by LrCas9, the results should be shown in the main figure instead of deletions (Fig. 2j, k). The text should also be modified.

Response: Thanks for your advice. Like SpCas9, LrCas9 introduces mainly 1-bp insertions and deletions of variable lengths. For this reason, we generated profiles of deletion positions and deletion lengths, as we did before for this kind of analyses (Ren et al., *Nat Plants* 7,25-33, 2021; Zhong et al., *Mol Plant* 12, 1027-1036, 2019; Tang et al., *Plant Biotechnol J* 17, 1431-1445, 2019).

Unify the presentation, e.g. sgRNA+PAM (Fig. 4a) and spacer PAM (Fig. 5a), etc.

Response: Thanks for point this out, and we unified it with “protospacer+PAM” on the final version.

Some of the sgRNA sequences in the figure are not identical, for example, OsTG01 (Fig. 4a, Fig. 5e, f) and OsTG02 (Fig. 6f, g). I would strongly recommend the authors to really carefully check entire manuscript and all the data.

Response: Thank you for your meticulous review of our work. After a thorough examination, we have identified and rectified the labeling discrepancies in Figure 5e, Figure 5f, Figure 6f, and Supplementary Figure 13 for the final version of the manuscript. Furthermore, we have meticulously revised the manuscript text to align with the corrected figures. Your feedback has been instrumental in enhancing the overall quality and accuracy of our research, and we thank you very much.

The results showed that the LrCas9-mediated CRISPRi suppressed the expression of target genes by ~20% or less (Fig. 7c). On the other hand, mutation efficiency in protoplasts was 20-50% (Fig. 2, 3). I am wondering why the LrCas9-mediated CRISPRi system is so efficient in protoplasts. As authors mentioned, rice protoplast transformation efficiency was ~90%. Does this mean that the suppression efficiency of the CRISPRi system via LrCas9 is almost 100%?

Response: Thanks for raising this point. Genome editing by Cas9 nuclease and NHEJ repair (which is pretty precise but can introduce errors due to repetitive Cas9 cutting) and CRISPRi are different systems, and the factors dictating their efficacy would be very different. For example, for genome editing, the mutated sites would typically be resistant to editing again, and hence the mutations can accumulate over time in the protoplast populations. For CRISPRi, it requires the binding of dCas9 protein to the promoter of the target gene. If the CRISPR components are well expressed in most cells (e.g., rice protoplasts with 90% or higher transformation efficiency), we can see significant mRNA reduction of the target gene, as we showed in our experiments. Anyway, we don't expect these different CRISPR applications would have similar efficiency. Thanks for your careful reading, and we are grateful for your suggestions to improve our work.